# Green Roofs: Nature-Based Solution or Forced Substitute for Biologically Active Areas? A Case Study of Lublin City, Poland

Malwina Michalik-Śnieżek *, Kamila Adamczyk-Mucha, Rozalia Sowisz and Alicja Bieske-Matejak

Department of Grassland and Landscape Planning, University of Life Sciences in Lublin, Akademicka 13, 20-950 Lublin, Poland; kamila.adamczyk@up.lublin.pl (K.A.-M.)
* Correspondence: malwina.sniezek@up.lublin.pl; Tel.: +48-81-445-60-30

**Abstract:** Green roofs have become an increasingly popular feature in building design, driven by their bio-physical properties and aesthetic and recreational values. They serve as a key element in promoting the integration of Nature-Based Solutions (NBSs) in urban fabrics, aiming to enhance urban environments, mitigate climate impact, and create more sustainable urban spaces. Polish regulations mandate that investors maintain a designated proportion of biologically active areas, ensuring natural vegetation and rainwater retention. Green roofs fulfill this requirement and can serve as compensation for the loss of biologically active areas due to construction. Unfortunately, the regulations lack specificity regarding their construction. This study aimed to examine whether green roofs consistently represent NBSs, as frequently presented in the scientific literature, or rather serve as a legal substitute for biologically active areas. The research was conducted in Lublin, the ninth largest city in Poland. Field studies, analysis of planning documentation, and review of administrative decisions have revealed that the majority of green roofs in Lublin have a greenwashing character, meaning they were installed to meet urbanistic indicators rather than for climate, environmental, or aesthetic reasons. Such studies have not been conducted before in relation to local spatial development plans and administrative decisions in Poland, and they show that this approach does not contribute to increasing biodiversity on investment plots. Notably, the investor would be denied construction permits without the incorporation of green roofs. Consequently, this leads to the conclusion that not all green roofs fulfill the criteria of NBS, as not all ensure an increase in biodiversity. Therefore, legal provisions regarding their establishment should be revised and specified.

**Keywords:** green roofs; Nature-Based Solutions (NBSs); biologically active areas; greenwashing; biologically active areas index; Poland

## 1. Introduction

The progressing and evident impacts of climate change adversely affect the well-being of inhabitants in large urban areas, as well as the development and expansion of urban areas themselves [1,2]. The evolving regulations regarding sustainable development necessitate the implementation of adaptive solutions in human living spaces to address the effects of climate change. Notably, Nature-Based Solutions (NBSs) [3] are considered among these adaptive measures. Green roofs, known as artificial ecosystems, exemplify complex structures and systematic approaches to addressing issues in modern urban areas. They perform diverse functions and provide benefits across various scales [4]. As previously mentioned by Mihalakakou G. and others [5], the benefits arising from the implementation of green roofs (GRs) can be classified into five groups. (1) Energy benefits related mainly to reducing the urban heat island (UHI) effect [6,7], where a significant factor is the type and quantity of plant material [8] as well as the actual percentage of plant surface coverage [9]. This is a crucial aspect for roofs beyond the warranty period or those entrusted to private care. Green roofs can influence both the immediate surroundings of the building and the thermal properties of the building itself.

Vera and others [10] observed that numerous studies focus on the impact of green roofs on the energy characteristics of the building, with few addressing the energy consumption of the entire building to investigate the impact of green roofs on the energy efficiency of the building. Undoubtedly, thermal insulation of the roof itself plays a vital role in reducing cooling and heating costs [11–14]. In the group of thermal factors, the Leaf Area Index (LAI) plays a significant role, influencing shading, evapotranspiration, and latent and convective heat fluxes [15,16], which have a substantial effect on the energy behavior of a system [7,17–19]. (2) Environmental benefits—primarily addressing air quality and water quality, green roofs extend vegetated spaces, positively impacting air quality by absorbing harmful pollutants and reducing $CO_2$ and $NO_2$ levels [20–22]. Slowing down rainwater runoff to the sewage system, decreasing the load on water treatment plants, and enhancing the quality of water runoff from the roof are crucial factors supporting the adoption of green roof solutions [23,24]. Multi-layered blue–green roofs provide an additional opportunity for rainwater storage during and immediately after heavy rains, allowing for remote control of their drainage [17]. To maximize water retention properties, it is essential to select a system tailored to specific climatic conditions. Research has been conducted in distinct climatic zones, such as the Mediterranean basin [25] and the Netherlands [26], where a 70% reduction in rainfall runoff has been investigated. Utilizing an appropriate system and species selection can increase the neutralization of acidic rainwater from a pH of 5–6 to 8 [18,27]. (3) Ecosystem benefits—within the realm of ecological benefits, highly tangible and visible is the enhancement of biodiversity through the creation of habitats beneficial to birds and insects [28–31]. A crucial factor influencing the presence of animals is the planned selection of species, including reliance on native, nectar-rich, and alternately blooming species [32,33].

Many researchers emphasize the role of native species and their selection for green roofs [18,19]. Emerging offerings of systemic biodiverse roofs are intriguing, serving as an alternative to extensive roofs [34]. Here, one can observe the need to strengthen the relationship between the research environment and the industry to create model solutions for increasing environmental benefits [35]. (4) Social–aesthetic and psychological benefits—green roofs play a crucial role in mitigating the adverse effects of climate change on both physical and mental well-being [36–39]. The correlation between improving urban air quality by increasing greenery and its positive impact on human health has been substantiated, as evidenced by studies conducted over 13 years in 95 urban communities [40]. The significance of greenery and direct access to it in urbanized spaces have gained immense importance, particularly after the COVID-19 pandemic. Green roofs can play a pivotal role in this context [41]. (5) Economic benefits (cost–benefits analyses)—the economic advantages of implementing green roofs, including the selection of technologies for both the roof and the building itself, necessitate meticulous cost–benefit analyses. This involves examining three key factors: initial costs, yearly cost-effectiveness, and overall efficiency [42,43]. An intriguing study conducted by Clark and colleagues [44] observed that the average construction cost of a green roof on a standalone residential building is 39% higher than that of a traditional roof. Over a 40-year building lifespan, the Net Present Value (NPV) analysis of a green roof is 20.3 to 25.2% lower than that of a conventional roof, accounting for reduced costs of rainwater fees and energy savings. When assessing financial benefits, choosing an appropriate method for Life-Cycle Analysis is essential considering the three stages of production, construction, and operation and encompassing energy consumption, greenhouse gas emissions, stormwater runoff, and rainwater utilization [43].

The proper classification of green roofs as NBSs is influenced by numerous factors. The integration of a green roof with suitable building technology, careful species selection to enhance biodiversity, low maintenance costs, an effective rainwater management system, and the physical or visual accessibility of the green roof for city residents all play pivotal roles. However, in the context of numerous investments and applications of green roofs, the phenomenon of 'greenwashing' is a common occurrence [45,46].

Implementation of green roof solutions in Poland is becoming increasingly popular. However, their presence is often a result of legal regulations that compel investors to maintain a specific proportion of 'biologically active area' within the boundaries of investment plots. According to the provisions of the regulation issued by the Minister of Development and Technology regarding the technical conditions that buildings and their locations should meet [47], the term 'biologically active area' refers to an area with a surface arranged to ensure natural plant vegetation and the retention of rainwater. It also includes 50% of the surface of terraces and flat roofs with such a surface, as well as other areas providing natural plant vegetation, with a minimum surface area of 10 m$^2$, and surface water on this terrain.

In spatial planning in Poland, until September 2023, the commonly utilized indicator for biologically active areas was expressed as the ratio of the surface area of biologically active lands to the building plot's surface area, represented as a percentage [48]. The determinations regarding the values of this indicator are present in the local spatial development plans of cities or municipalities. In the absence of such plans, they can be found in administrative decisions on building and land development conditions issued by city or municipal authorities. Since the autumn of 2023, new regulations have been in force, mandating that biologically active areas must be covered with a surface ensuring natural plant vegetation and rainwater retention (not necessarily native soil, as implied by previous definitions). This suggests that 100% of roof gardens with such surfaces may constitute biologically active areas.

Such a solution, in which the participation of biologically active areas is determined by the possibility of natural plant vegetation and rainwater retention, rather than compensation for biodiversity lost due to investment, resulted in the clearance of old trees on small investment plots. To maintain an appropriate share of biologically active areas, sedum roofs, a type of extensive green roof, were established. Their construction is less expensive due to the materials used and their quantity compared to more intensive roof types. Therefore, it is worth asking whether the promotion of green roofs as NBSs, without indicating their harmful effects on urban biodiversity conservation, should be implemented.

The aim of the presented research was to verify green roofs in Lublin in terms of their purpose, i.e., whether their establishment was guided by the ideology related to creating NBSs for climate change adaptation, achieving the appropriate values of the urban indicator for biologically active areas, or other premises. The goal was also to check whether green roofs in the city can actually be considered NBSs, following the EU definition [3].

In this article, it has been demonstrated that in the case of the majority of investments involving green roofs in Lublin, their inclusion in the investment project aimed at achieving a biologically active area index, the value of which was required by provisions of the local spatial development plan or administrative decision on development conditions. The majority of the examined extensive green roofs are difficult to interpret as NBSs due to their construction and quality, as they do not provide an increase in biodiversity compared to adjacent areas.

## 2. Materials and Methods

### 2.1. Study Area

Research was conducted in the area of the city of Lublin, the capital of the Lublin Voivodeship, located in eastern Poland. With a population exceeding 310,000 [49], it ranks as the 8th largest city in Poland. According to the diagnosis conducted for the development of the city's climate change adaptation plan [50], the area within the city's borders is in a zone of very high risk of high temperatures, drought, and hot days, as well as a very high risk of sudden floods and urban floods. The overall share of biologically active surface areas in the city is 64% [51], but this is a result of the distribution of large forest complexes and agricultural fields on the city's outskirts. In the downtown area, permeable areas account for only 40% compared to hardened and built-up areas. In the zone of compact urban development, biologically active areas are displaced by multifamily and

office buildings (Figure 1). Lublin is a city where biodiversity is decreasing due to economic investments, road infrastructure modernization, residential construction development, and the construction of large-scale commercial facilities [52]. The city's ecological system, as outlined in planning documents [53], comprises three valleys, regulated rivers, a retention reservoir, urban forests, and numerous dry valleys. The establishment of this system aimed to counteract environmental degradation and preserve its values, yet it proved inadequate for maintaining the city's biodiversity. The city's ecological system lacks coherence, and it is partially owned by private individuals who have the right to manage their land. The decline in biodiversity is most evident in the central part of the city. According to the eco-physiographic study conducted in Lublin's city center [54], actual vegetation significantly differs from potential, with ruderal vegetation dominating, while green areas are mainly characterized by lawns with individual tree and shrub plantings. Although there are parks and urban squares in the city center, they are not connected by corridors.

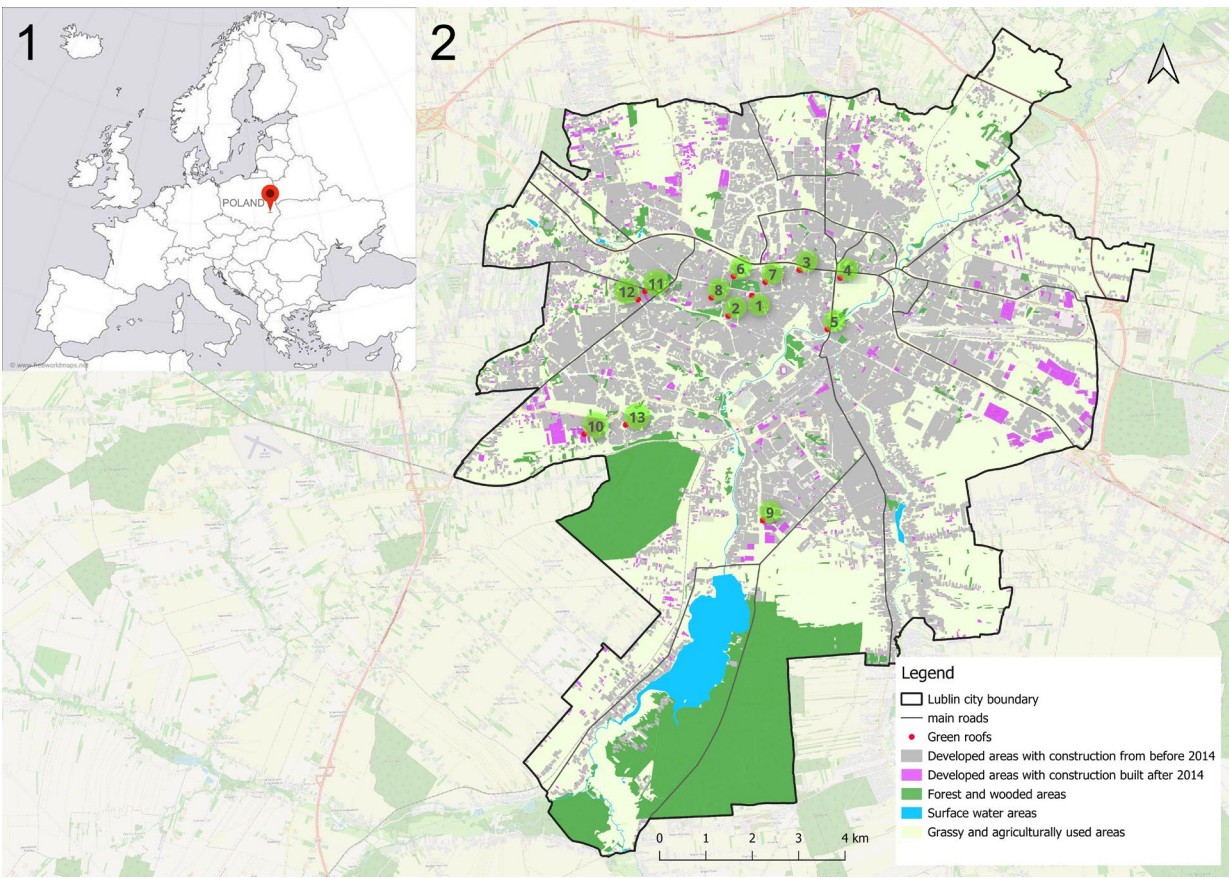

**Figure 1.** The geographical location of the city of Lublin (**1**) in relation to European and Polish administrative boundaries; (**2**) green areas (biologically active surfaces) and built-up areas within the city limits [50]. Numbers 1–13 denote the positions of green roofs in accordance with the sequence outlined in Table 1.

*2.2. Selection of Research Objects*

This study focused on residential, commercial, and public utility buildings within the boundaries of the city of Lublin. The selection criteria included buildings where the top floor or terrace area facilitates natural water retention processes and has been covered with living vegetation. Underground parking ceilings, where vegetation is at ground level, were excluded from this study. To locate buildings meeting these criteria, an analysis was conducted using data provided by the City of Lublin available on the municipal geoportal [55]. This dataset includes 2022 orthophotomaps with a spatial resolution of 0.10 px and geodetic information containing the distribution, outline, and number of floors

of the buildings. Additionally, information from the Department of Architecture and Construction of the City of Lublin regarding the location of investments featuring green terraces or roofs was considered. The obtained information was verified based on data found on the websites of development companies utilizing these solutions for marketing purposes and institutions implementing investments incorporating green roofs. The objects subject to examination have been compiled in Table 1, and their locations are accessible at the address: https://storymaps.arcgis.com/stories/4f63a83d47e2479eb682b82eddeb47d6 (accessed on 20 December 2023).

**Table 1.** The list of buildings with green roofs in Lublin, indicating the type of green roof applied in project implementation.

| No. | Building Name | Project's Completion Date | Characteristics | Type of Green Roof |
|---|---|---|---|---|
| 1 | Voivodeship Cultural Center 'Centrum Spotkania Kultur' | 2015 | Building is a public facility managed by the Lublin Voivodeship self-government. It functions as a theater, philharmonic, art gallery, and conference center, featuring seven above-ground floors with a total area of 30,000 square meters. | Intensive * and extensive ** |
| 2 | Library of the University of Life Sciences in Lublin | 2012 | The building is a public facility affiliated with the University of Life Sciences in Lublin, serving as the Regional Center for Agricultural Scientific Information. Its primary function is as a library, and it also serves as a conference facility. It features six above-ground floors with a total area of 4300 square meters. | Semi-intensive *** |
| 3 | Public clinical hospital No. 1 | 2023 | The public building is owned by University Clinical Hospital No. 1 in Lublin. It accommodates medical care units. It features eight above-ground floors with a total area of 15,000 square meters. | Extensive |
| 4 | Shopping Center 'Vivo!' | 2015 | A commercial, retail, and service building with one above-ground floor, located in the center of Lublin near Lublin Castle; it covers an area of 104,000 square meters. | Intensive and extensive |
| 5 | 'Arche' Hotel | 2018 | Commercial building, hotel, four above-ground floors, and one underground; area of 5000 square meters. | Extensive |
| 6 | 'Centrum Park' office/apartment building | 2015 | Private building consisting of two parts: office–service and residential. The building has 8 above-ground floors. | Extensive |
| 7 | 'Spokojna 2' office building | 2018 | Private, commercial building, with 7 above-ground floors. Area: 35,000 square meters. | Extensive |
| 8 | 'Bema 1' apartment building | 2001 | Private residential building with 6 above-ground floors. | Intensive |
| 9 | 'Sky Gardens' apartment building | 2015 | Residential building located at Domeyki Street, private, with 6 above-ground floors and 150 apartments. | Semi-intensive |
| 10 | 'Forest Retrit' apartment building | 2017 | Residential building, multifamily, consisting of two parts: 4 floors and 5 floors. | Extensive |
| 11 | 'Studio Residance' apartment building | 2020 | Seven-story multifamily building with commercial function. | Extensive |

**Table 1.** *Cont.*

| No. | Building Name | Project's Completion Date | Characteristics | Type of Green Roof |
|-----|---------------|--------------------------|-----------------|--------------------|
| 12 | 'Wojciechowska 5' apartment building | 2020 | Seven-story multifamily building with commercial function. | Extensive |
| 13 | 'Szafirowa 7' apartment building | 2019 | Multifamily housing complex consisting of 2 buildings, 4 and 5 stories high. | Extensive |

\* Intensive GRs have deep substrates (>25 cm) and a wide variety of vegetation that can include shrubs and trees [56,57]; \*\* Extensive GRs are shallow, light-weight systems (60 to 150 kg/m$^2$) that typically have a growing medium depth of 5 to 15 cm [58]. Plant diversity is generally limited due to the shallow substrate depths; \*\*\* Semi-intensive GRs generally accommodate a wide variety of types of vegetation due to their substrate depth. Intensive GRs have deep substrates (>25 cm) and a wide variety of vegetation that can include shrubs and trees [56,57].

*2.3. Verification of Green Roof Areas as Components of the Biologically Active Areas Index*

To ascertain the role played by the implementation of a green roof within a specific construction project, the initial step involved determining its contribution to achieving the required biologically active areas index. As part of this study, existing green roofs within the city were analyzed, referring to the regulations in force at the time of the project's completion.

By scrutinizing the provisions of local spatial development plans [59] and decisions on building and land development conditions (provided by the Lublin City authorities), the required values of this index were verified. Subsequently, calculations were performed to assess how this index would be shaped if the project did not incorporate a green roof.

To calculate the current biologically active areas index for a given investment, both with and without considering the green roof, a formula was applied, which was developed based on the definition of a biologically active area contained in the Regulation of the (Polish) Minister of Infrastructure of 12 April 2002, regarding the technical conditions to be met by buildings and their location, which states that a 'biologically active area is an area with a surface arranged in a way that ensures natural plant vegetation and retention of rainwater, as well as 50% of the surface of terraces and flat roofs with such a surface and other surfaces providing natural plant vegetation, with an area of not less than 10 m$^2$, and surface water on this land [47].' Because in local spatial development plans for Polish cities and municipalities the required value of a biologically active land area is given in percentage and applies to all cadastral parcels on which the investment is implemented, the formula looks as follows:

$$Bai = \frac{\left(Pa - BP + \frac{1}{2}Gr\right)}{Pa} \times 100\%$$

*Bai*—biologically active areas index (%);
*Pa*—plot area (the sum of the areas of the cadastral parcels of the investment (m$^2$);
*BP*—build-up and paved area (m$^2$);
*Gr*—green roof area, not less than 10 m$^2$ (m$^2$).

The area of investment parcels containing buildings with green roofs, the area of built-up (hardened) areas, as well as the area of the green roofs themselves, were determined based on geodetic dataset (https://geoportal.lublin.eu/2d/, accessed on 20 December 2023).

*2.4. Verification of Green Roofs as an NBS*

The next stage of the research involved examining which green roofs in Lublin meet the criteria enabling their classification and consideration as NBSs. In accordance with the results of analyses of numerous definitions and studies conducted by Sowińska-Świerkosz and Garcia [60], for solutions to be considered as NBSs in space, they must fulfill the following conditions: (1) are inspired and powered by nature; (2) address (societal) challenges or resolve problems; (3) provide multiple services/benefits, including biodiversity gain; and (4) are of high effectiveness and economic efficiency. This means that failure to meet any of

these conditions disqualifies a solution as an NBS. Lists of features qualifying solutions as NBSs, as presented in Figure 2, were employed to achieve this goal.

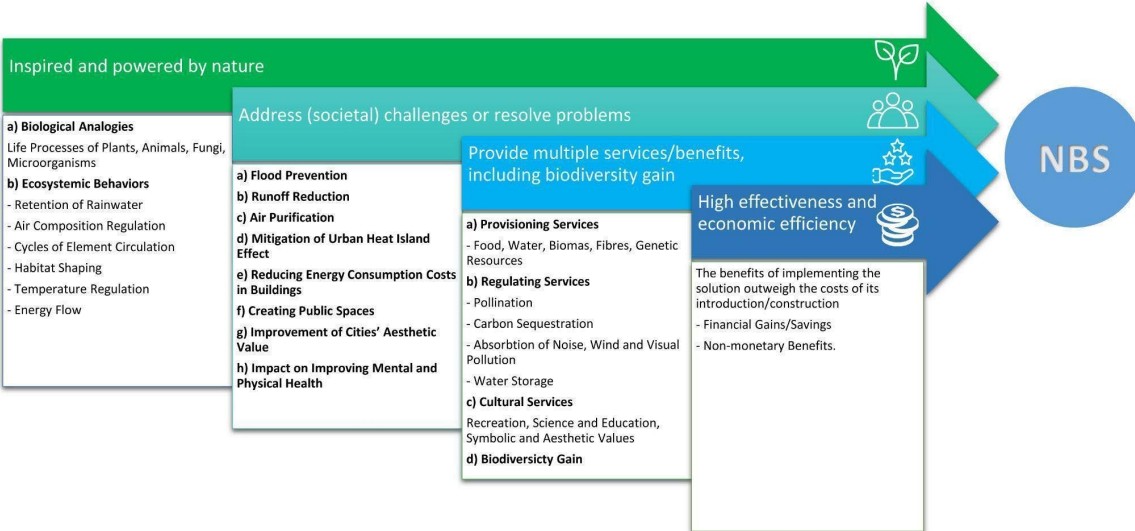

**Figure 2.** Lists of features for each criterion that must be met to classify a solution as an NBS [52,55].

The coverage of a building's roof with living vegetation, whose life processes are sustained by natural phenomena (such as precipitation, sunlight, and processes occurring in the soil or substrate), along with the effects of these processes (e.g., air purification, runoff reduction) and the benefits derived from them (Figure 2), suggests the classification of such solutions as NBSs. Similarly, the economic efficiency of these assumptions qualifies green roofs as NBSs, as evidenced by ongoing scientific research that allows for the improvement of the energy efficiency of buildings and the reduction of emissions from heating and cooling processes [14], carbon dioxide absorption [61], and contributions to savings in rainwater drainage costs and other savings [62]. However, when verifying green roofs as NBSs, it is necessary to consider the criterion of biodiversity growth. Polish law regulations do not require investors to ensure a net increase in biodiversity, as is the case in the United Kingdom's law [63]. To determine this growth or its absence, methods developed for the Biodiversity Metric 3.1 were applied [64]. This metric utilizes the following criteria: (1) distinctiveness, (2) condition, and (3) strategic significance.

Distinctiveness in the case of green roofs refers to the uniqueness of artificially created habitats in relation to habitats in the surrounding area. The evaluation of this criterion for green roofs was carried out using a binary method by assigning a value of 0 when the habitat has the same characteristics as habitats within the same cadastral plot but at the ground level or adjacent cadastral plots and 1 in the case of habitats with different characteristics than those in the vicinity. Sedum matting, with a thickness of up to 60 mm, is characterized by a low value as compensation for biological diversity, and in the case of the distinctiveness assessment, it achieves low scores [65]. An exception may be the case of a lack of natural habitats nearby. In such a case, even sedum matting with a small thickness can contribute to the increase in biodiversity.

The condition criterion for green roofs was also assessed in a binary fashion. A value of 1 was assigned to solutions where the health and quality of plants were assessed as equal to those in healthy natural habitats, while 0 was assigned when the quality was assessed as lower than in healthy natural habitats. Assessments were made using an expert method applying plant biology and plant community research methods [66,67].

The criterion of strategic significance could not be assessed considering the provisions of the Biodiversity Metric 3.1 due to a lack of necessary data. However, an attempt was made to assess strategic significance through the prism of the role in the Green and Blue Infrastructure (GBI) network. Green roofs, the presence of which caused an increase in the

share of biologically active areas of investment plots by more than 5%, received a value of 1, while those whose presence did not lead to a significant increase in the share of biologically active areas (5% or less) received a value of 0.

Green roofs that received 3 points in this assessment were classified as those that provide an increase in biodiversity and thus meet all of the necessary criteria for their classification as NBSs.

## 3. Results

The conducted analyses revealed that in the city of Lublin, with a total area of 147,400,000 m$^2$, as of June 2023, only 13 buildings were equipped with green roofs. The total green roof area in the city is 20,767 m$^2$, which represents a mere 0.014% of the city's total area. Among them, three are public utility buildings, four are service buildings, and six are residential buildings. The largest green roof area is found on a commercial building housing a shopping center. Out of them, eight buildings exclusively feature extensive roofs, two have mixed roofs, and two have semi-intensive roofs (Table 1). Examples of green roofs in Lublin are presented in Figure 3.

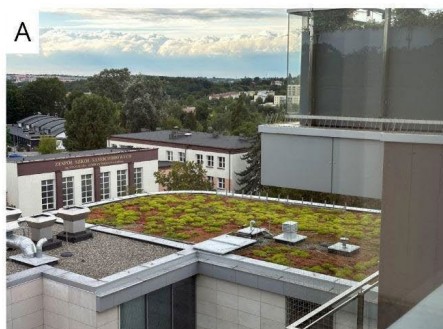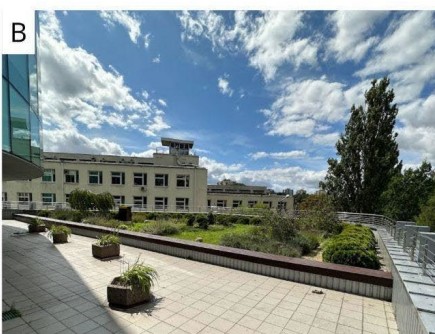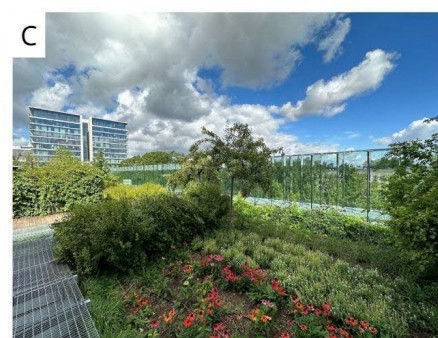

**Figure 3.** Examples of green roofs present in the city of Lublin. (**A**) Extensive green roof on a commercial building, the Central Park Office Building. (**B**) Semi-intensive green roof on the Library of the University of Life Sciences in Lublin. (**C**) Intensive roof at the Voivodeship Cultural Center 'Centrum Spotkania Kultur'.

### 3.1. Verification of Green Roof Surfaces as Components of the Biologically Active Areas Index

The required value of the biologically active areas index for developments incorporating green roofs in Lublin varies from 15% to 35%. Most developments would not meet the required value of this index if it were not for the implementation of a green roof project. This is particularly relevant for residential investments undertaken by private investors. It can be presumed that local regulations and conditions of development and land use compelled investors to construct a green roof as part of the construction project (Table 2 and Figure 4). An exception is the investment involving a green roof on Clinical Hospital No. 1 on Staszic Street. In this case, the required index value was 15% (decision on development and land use conditions), but the achieved index value was 27%. According to the regulation [47], for healthcare buildings, the biologically active area index should be at least 25%. Therefore, considering the overall index value, the green roof also contributed to its attainment. An interesting example is the 'Sky Gardens' apartment building on Domeyka Street. In this case, the required index determined by the decision on development conditions would have been achieved even without green roof landscaping, but the name of the apartment building itself suggests that this action had a strictly marketing-oriented purpose.

**Table 2.** Required and achieved biologically active area index values for investment plots.

| No. | Building Name | Investment Plot Area/GR Area (m²) | $B_{ai}$ Excluding GRs Area | $B_{ai}$ Including GRs Area | $B_{ai}$ Required by Law or Agreements |
|---|---|---|---|---|---|
| 1 | Voivodeship Cultural Center 'Centrum Spotkania Kultur' | 12,762/ 2488 | 13% | 22% | Not specified |
| 2 | Library of the University of Life Sciences in Lublin | 22,155/ 157 | 21.6% | 22% | Not specified |
| 3 | Public clinical hospital No. 1 | 30,252/ 1340 | 23% | 27% | 15% (AD *) |
| 4 | Shopping Center 'Vivo!' | 29,204/ 16,700 | 5.7% | 34% | 26% (plan **) |
| 5 | 'Arche' Hotel | 2631/ 655 | 2% | 15% | 15% (AD) |
| 6 | 'Centrum Park' office/apartment building | 3810/ 355 | 20% | 25% | 25% (AD) |
| 7 | 'Spokojna 2' office building | 6125/ 832 | 9% | 15% | 15% (AD) |
| 8 | 'Bema 1' apartment building | 1796/ 200 | 30% | 41% | Not specified |
| 9 | 'Sky Gardens' apartment building | 6837/ 1300 | 25.6% | 34.6% | 25% (AD) |
| 10 | 'Forest Retrit' apartment building | 9157/ 350 | 23% | 25% | 25% (plan) |
| 11 | 'Studio Residance' apartment building | 5634/ 600 | 20% | 25% | 25% (plan) |
| 12 | 'Wojciechowska 5' apartment building | 4980/ 850 | 17% | 25% | 25% (plan) |
| 13 | 'Szafirowa 7' apartment building | 8780/ 970 | 32% | 37% | 35% (plan) |

* AD—according to the administrative decision on building and land development conditions; ** plan—according to the local development plan.

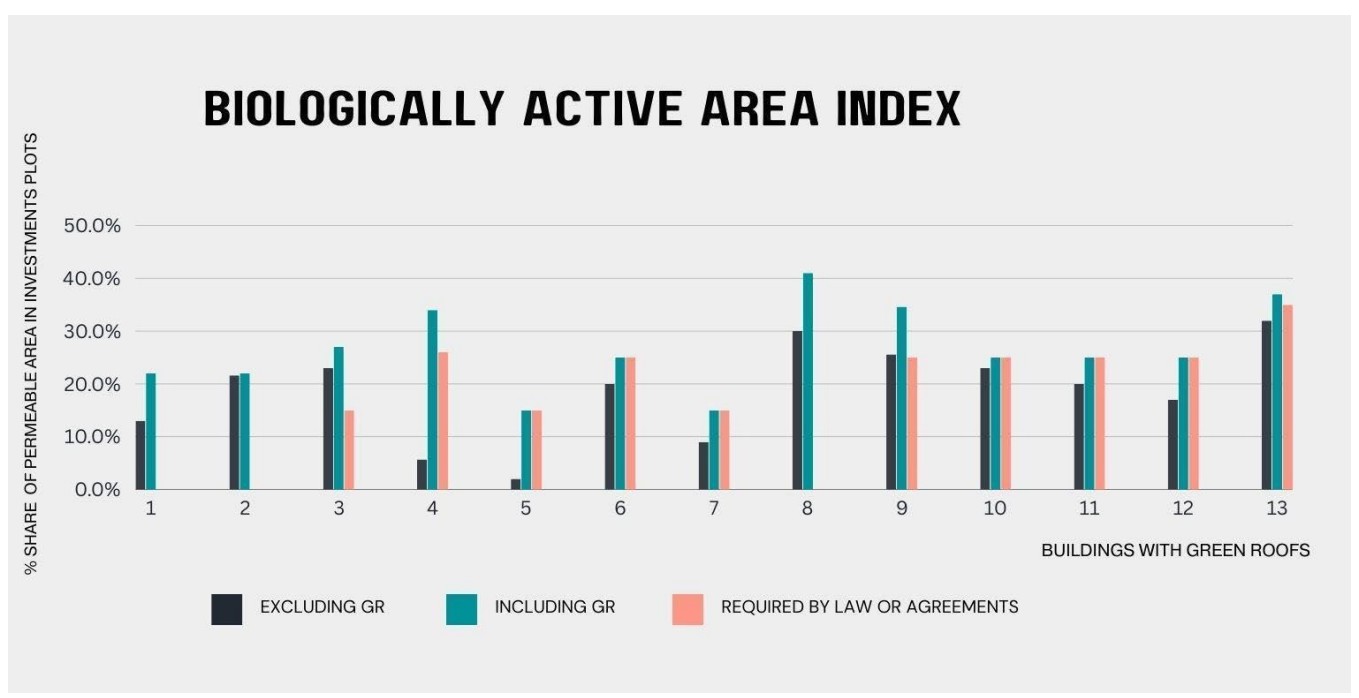

**Figure 4.** The required and achieved values of the biologically active area index for individual construction projects incorporating green roofs (building numbers correspond to the ordinal numbers in Table 1).

### 3.2. Verification of the Examined Green Roofs in Terms of NBS Classification Criteria

The conducted inventory work revealed that out of 13 green roofs in Lublin, 8 are characterized by the presence of an extensive roof with a substrate thickness not exceeding 6 cm covered with sedums (usually one species in several varieties) having a ground-covering nature. The vegetation on these roofs is not subject to maintenance activities and is largely imperceptible to residents or building users, or can only be observed to a very limited extent (e.g., in the case of building No. 3, only patients of certain wards whose rooms are located in specific places can observe the vegetation on the roof). In the case of three roofs, the substrate was characterized by a thickness of 15–25 cm, and these roofs are usually covered with medium-sized plants, mainly ornamental shrubs, indicating their aesthetic use. Observations and interviews with building managers indicate that they undergo only necessary maintenance work. The number of species used for planting is diverse, but it usually does not exceed 10–15. However, in the case of the roof of building No. 9, the number of species is higher due to the roof functioning as residents' 'backyard gardens.' These are small quantities of species compared to grassland communities occurring in the vicinity of some of these premises, where the floristic composition may include up to 230 taxa [68]. Two buildings have been equipped with intensive green roofs characterized by a significant substrate depth, diverse types of vegetation, and numerous species. These roofs are located in areas with low biodiversity resulting from the presence of dense buildings, paved transport routes, and high soil pollution. Their presence thus constitutes an important ecological node in the city (Table 3).

**Table 3.** GR specifications for buildings in Lublin.

| No. | Building Name | Substrate Thickness | Vegetation Types | Plant Species Count | Plant Height | Maintenance |
|---|---|---|---|---|---|---|
| 1 | Voivodeship Cultural Center 'Centrum Spotkania Kultur' | >50 cm (intensive) 6–10 cm (extensive) | Mosses, perennials (sedum), grasses, annual plants, shrubs, trees | 42 [69] | Low, medium, and tall | Full range of maintenance works including watering and fertilizing |
| 2 | Library of the University of Life Sciences in Lublin | 15–25 cm | perennials, grasses, annual plants, shrubs | 15 [70] | Low, medium | Full range of maintenance works including watering and fertilizing |
| 3 | Public clinical hospital No. 1 | 4–6 cm | Sedum | One species in varieties | Low | No maintenance |
| 4 | Shopping Center 'Vivo!' | >50 cm (intensive) 6–10 cm (extensive) | Mosses, perennials (sedum), grasses, annual plants, shrubs, trees | >50 | Low, medium, and tall | Full range of maintenance works including watering and fertilizing |
| 5 | 'Arche' Hotel | 4–6 cm | Sedum | One species in varieties | Low | No maintenance |
| 6 | 'Centrum Park' office/apartment building | 4–6 cm | Sedum | One species in varieties | Low | No maintenance |
| 7 | 'Spokojna 2' office building | 4–6 cm | Sedum, grasses | 5–10 | Low | Periodic watering and trimming of lawns |
| 8 | 'Bema 1' apartment building | 15–20 cm | Grasses, shrubs | 5–10 | Low, medium | Periodic watering and trimming of lawns |
| 9 | 'Sky Gardens' apartment building | 15–25 cm | Perennials, grasses, annual plants, shrubs | 30–50 | Low, medium | Diverse maintenance tailored to the owner's preferences |
| 10 | 'Forest Retrit' apartment building | 4–6 cm | Sedum | One species in varieties | Low | No maintenance |
| 11 | 'Studio Residance' apartment building | 4–6 cm | Sedum | One species in varieties | Low | No maintenance |
| 12 | 'Wojciechowska 5' apartment building | 4–6 cm | Sedum | One species in varieties | Low | No maintenance |
| 13 | 'Szafirowa 7' apartment building | 4–6 cm | Sedum | One species in varieties | Low | No maintenance |

The assessment of the examined GRs from the perspective of ensuring biodiversity gain revealed that the criterion of distinctiveness of artificially created habitats is met by three roofs. Two of them are roofs where both intensive and extensive parts are present, and one is a semi-intensive roof. In the case of Roof No. 1, the intensive part contains deep substrate and a diverse range of plant species, including the presence of tall trees, including fruit-bearing ones. However, the extensive part consists of sedum roofs with low species diversity. Consequently, the intensive part received a positive evaluation regarding the distinctiveness criterion, while the extensive part did not.

For GR No. 4, located in the 'Vivo' Shopping Center, both the intensive and extensive parts received a positive evaluation for distinctiveness. Both the intensive and extensive parts of the roof contain a diverse range of plant species, and the large surface of these installations serves as habitats for other organisms. The semi-intensive roof No. 9, 'Sky Gardens,' on a residential building, also received a positive assessment; despite having a shallower substrate compared to previous roofs, it contains a diverse range of plant species.

The remaining roofs are mainly of the extensive type, specifically sedum roofs with a thickness of up to 60 mm, where the species composition includes no more than 10 plant species and a few insect species. The habitats of sedum roofs are not exceptional or unique compared to habitats in the vicinity, both within the same cadastral plot and neighboring plots.

In the assessment of the habitat conditions of GRs on the examined buildings, as many as nine of them exhibit good condition, including both intensive and some extensive roofs. It was observed that the condition of extensive green roofs decreases with their age, with older buildings having poorer evaluations than those constructed later. This may indicate a lack of proper maintenance or low-quality substrate used for establishing the green roofs, which over time ceased to fulfill their function adequately.

Green roofs on seven buildings received a point for strategic significance, with the greatest increase in biologically active surface provided by the green roof on building No. 4, the Shopping Center 'Vivo!'

Analyzing the obtained results, the implementation of NBSs is fulfilled by roofs on three buildings: (1) the Voivodeship Cultural Center 'Centrum Spotkania Kultur'; (2) the Shopping Center 'Vivo!'; and (3) the 'Sky Gardens' apartment building (Table 4).

**Table 4.** The evaluation of green roofs (GRs) from the perspective of ensuring biodiversity growth.

| No. | Building Name | Distinctiveness | Condition | Strategic Significance | Total Points |
|-----|---------------|-----------------|-----------|------------------------|--------------|
| 1 | Voivodeship Cultural Center 'Centrum Spotkania Kultur' | 1 (intensive)<br>0 (extensive) | 1<br>1 | 1<br>1 | 3<br>2 |
| 2 | Library of the University of Life Sciences in Lublin | 1 | 1 | 0 | 2 |
| 3 | Public clinical hospital No. 1 | 0 | 1 | 0 | 2 |
| 4 | Shopping Center 'Vivo!' | 1 (intensive)<br>1 (extensive) | 1<br>1 | 1<br>1 | 3<br>3 |
| 5 | 'Arche' Hotel | 0 | 1 | 1 | 2 |
| 6 | 'Centrum Park' office/apartment building | 0 | 0 | 0 | 0 |
| 7 | 'Spokojna 2' office building | 0 | 1 | 1 | 2 |
| 8 | 'Bema 1' apartment building | 0 | 0 | 1 | 1 |
| 9 | 'Sky Gardens' apartment building | 1 | 1 | 1 | 3 |
| 10 | 'Forest Retrit' apartment building | 0 | 0 | 0 | 1 |
| 11 | 'Studio Residance' apartment building | 0 | 1 | 0 | 1 |
| 12 | 'Wojciechowska 5' apartment building | 0 | 1 | 1 | 2 |
| 13 | 'Szafirowa 7' apartment building | 0 | 1 | 0 | 1 |

## 4. Discussion

Green roofs can be a significant element of sustainable urban development, providing healthy spaces for residents and mitigating adverse effects of climate change [63,64]. However, it turns out that green roofs can also serve as a form of greenwashing [46,71–74]. In the Polish context, greenwashing can be defined as a strategy employed by investors who replace biologically active areas present on native soil before the development of an investment with green roofs using sedum mats, the thickness of which is minimal for plant growth, including, typically, sedums. Succulents are relatively resilient and characterized by low water consumption. However, compared to grasses and other perennials, they do not provide the highest retention or cooling of rainwater [75]. The quantity of green roofs in other European countries is incomparably greater than in the surveyed area, the city of Lublin. This is accompanied by a heightened awareness of the benefits derived from establishing such surfaces in various spheres: energy, environmental, social, aesthetic, ecological, and economic. The latter are distributed either over time or directly through specific subsidies and incentives resulting from investment realization.

Another factor is the legal regulations arising from the necessity of implementing green roofs in newly developed projects. According to Burszta-Adamiak and others [76], eight European countries employ direct financial benefits for the implementation of green roofs, such as property tax breaks or incentives for rainwater drainage to rainwater sewers and financial subsidies for investment implementation. An interesting financial solution included in the Hamburg Strategy, implementing the principles of the green smart city concept, involves tax breaks up to 50%, co-financing of roof implementation up to 60%, and a reduction in fees or complete exemption from fees for draining rainwater to rainwater sewers.

Similar financial support for project implementation costs for private houses, apartment buildings, and offices is possible in the Netherlands, Switzerland, and Belgium. Noteworthy are the green roofs in Vienna, Austria, where, in addition to subsidies for project implementation, support is offered for maintenance and care, monitored by inspection twice a year.

Financial incentives are accompanied by legal solutions that mandate the implementation of green roofs in both new and revitalized buildings. For example, in France, newly constructed buildings since 2015 must have green roofs or photovoltaic panels in commercial districts. In Copenhagen, Denmark, all newly constructed buildings with a roof slope less than 30° must have green roofs. Similar regulations have been adopted in Germany for cities like Essen, Munich, or Esslingen, Stuttgart.

From the perspective of comprehensive support not only for biologically active surfaces but also for their high quality in terms of species biodiversity, the legal regulation adopted for Berlin, the Biofactor–Biotop Area Factor (BAF), is noteworthy. This regulation could successfully serve as guidance for the development of legal regulations as a tool for truly improving the quality of biologically active surfaces while simultaneously maintaining and developing the residential and commercial functions of the city.

A similar assumption, which also assumes very high quality of biologically active surfaces, is the Biofactor–Green Space Factor (GSF) adopted for Malmo, Sweden. Such regulations would undoubtedly change the quality of already existing green roofs in Lublin and increase their quantity through localization in newly emerging and planned developments.

Currently, when analyzing the existing green roofs in Lublin, it is clear that not all of them can be classified as NBSs, as they do not serve as compensation for biodiversity loss in the environment after investment realization. The absence of biodiversity maps for Lublin hinders the enforcement of regulations on investors concerning the circulation of elements, habitat protection, and shaping. Legislative models, such as BAF or GSF, would ensure the implementation of green roofs that enhance biodiversity not only in the context of statutory biologically active surfaces.

The preceding studies indicate that the biologically active area indicator alone is insufficient for enhancing biodiversity quality following investment completion.

The benefits of implementing green roofs, as outlined by Mihalakakou et al. [3], have been categorized into five groups: energy benefits, environmental benefits, ecosystem benefits, social–aesthetic and psychological benefits, and economic benefits (cost–benefit analysis). Examination of all completed roofs in Lublin (refer to Table 3) reveals that certain anticipated benefits from the aforementioned categories are not realized or expected by investors. In many instances, the pivotal factor is the biologically active surface area achieved on the plot (Table 2). To fully realize benefits, multiple aspects must be considered, with the operational lifespan of the building being a significant determinant. Research indicates deficiencies at the outset in assessing the quality of the green roof solution and the actual benefits emerging post-implementation over time. Energy and economic benefits (cost–benefit analyses) may accrue over an extended period of use [9–12]. The research underscores the absence of administrative mechanisms in Lublin for evaluating the green roof's quality over time, thus directly impacting the attainment of these two types of benefits. The analysis (Table 1) reveals lower greenery quality and inadequate maintenance, potentially impeding the attainment of energy and economic benefits.

The Leaf Area Index (LAI), an essential factor closely intertwined with thermal considerations [13,14], is disregarded in local administration or regulatory frameworks. The analysis indicates a lack of tools for assessing the quality and maintenance of greenery, as well as the LAI factor, indirectly linked to biodiversity and ecosystem benefits [26–31].

Social–aesthetic and psychological benefits stem from visual exposure and accessibility for visits [34–37]. Eleven out of the thirteen tested roofs partially fulfill this criterion, while two fully comply. The latter enable roof observation from within, are open to visitors, and are visible from neighboring structures. Additionally, in these two instances, green roofs serve as prominent image elements of the investment, including educational aspects focused on biodiversity. Figure 3 illustrates photographs of roofs from within the corresponding buildings.

There is a lack of biodiversity maps for the Lublin area, so research on the loss or preservation or improvement of biodiversity can be carried out based on the existing biodiversity of adjacent plots. Lublin lacks financial support for investors who have implemented green roofs (GRs). Despite this, such projects lead to numerous future benefits. These advantages include improved biodiversity and positive social impacts. Since the autumn of 2023, new law regulations have indicated that biologically active areas must be covered with a surface ensuring natural plant vegetation and rainwater retention. So, all of the green roofs can be classified as biologically active areas, but there is no tool to verify the quality of the greenery, especially after a period of time from the moment of putting into operation the investment. The very small number of green roofs in Lublin (Table 4, No. 13) compared to other European cities (e.g., Copenhagen, Denmark, where all newly constructed buildings with a roof slope less than 30° must have green roofs [66]) highlights issues or the lack of administrative tools to encourage investors to utilize green roofs as a means of effectively combating climate change. The results of the research show that 4 out of 13 green roofs in Lublin can be classified as NBSs, indicating the need to implement tools to achieve a higher quality of biologically active areas. An element that could significantly improve the number of green roofs in Lublin could be the use of existing models in other countries, such as legal regulations and incentive programs [76]. An increased number of green roofs in the city could be reflected in a real increase in biodiversity after the investment is completed; aesthetic value; counteracting climate change; and financial benefits (reliefs or support for investments using GRs and long-term investments related to the energy efficiency of the building during its operation).

## 5. Conclusions

The presented research, based on analyses of green roofs covering buildings constructed until 2022 in the study area, Lublin, indicates that such construction solutions were not popular at that time, as they constituted only a small fraction of roof coverings.

Analysis of legal provisions and administrative decisions related to the construction of investments (buildings with green roofs) revealed that in eight cases, the installation of a green roof was necessary to achieve the indicator of a biologically active surface. Without meeting this indicator, the realization of the investment would not be possible (the investor would not obtain a building permit). Furthermore, in all cases, these were commercial buildings (apartment buildings, hotels, office buildings, and shopping centers).

The majority of green roof coverings (8 out of 13) were extensive roofs, with sedum matting and a thin substrate (up to 6 cm), while only 2 were intensive roofs featuring large trees. This may be due to the fact that the presence of a green roof is checked at the technical acceptance of the building, and its presence in subsequent times is not verified. This is also supported by the condition of extensive green roofs, which significantly declines with the size of the building on which such a green roof is constructed.

Not all green roofs in Lublin can be considered Nature-Based Solutions (NBSs), as not all meet the condition of preserving biodiversity growth. Extensive green roofs with sedum matting and a poor species composition, as well as a thin substrate, do not ensure biodiversity growth compared to biologically active surfaces located at ground level in the immediate vicinity of buildings with green roofs.

In the case of Lublin, two green roofs of an intensive nature, one semi-intensive roof, and one extensive roof can be considered NBSs.

Green roofs may be considered greenwashing if their inclusion in a construction project is not primarily aimed at environmental and climate protection but rather at achieving the percentage of biologically active areas required by local laws or administrative decisions. The introduction of green roofs on buildings that cannot serve as NBSs is typically not due to ill intentions of the authorities but rather stems from flawed legal provisions that focus on quantitative control of biologically active areas rather than their quality and biodiversity.

General conclusions:

1. The research findings reveal a limited number of implemented green roofs in Lublin, with only 13 identified in total.
2. Among these green roofs, the majority are classified as extensive, with only two categorized as intensive.
3. The presence of green roofs on hotel and apartment buildings seems to be primarily motivated by regulatory compliance rather than environmental considerations.
4. Inadequate maintenance seems to compromise the quality of green roofs, especially on older buildings. This highlights the need for quality verification tools for biodiversity surfaces.
5. A lack of financial support and incentive programs for green roof implementation may contribute to the limited adoption of this technology in Lublin.
6. Despite the environmental benefits often associated with green roofs, not all implemented structures meet the criteria for Nature-Based Solutions, which indicates a need for further assessment and improvement.
7. The lack of tools to assess demand for biologically active areas shows a gap in local regulations. This underscores the need for policy changes to encourage the adoption of green roofs and ensure their effectiveness in reducing environmental impacts.

**Author Contributions:** Conceptualization, M.M.-Ś.; methodology, M.M.-Ś.; software, M.M.-Ś.; validation, M.M.-Ś.; formal analysis, M.M.-Ś. and K.A.-M.; investigation, M.M.-Ś. and K.A.-M.; resources, M.M.-Ś. and K.A.-M.; data curation, M.M.-Ś. and K.A.-M.; writing—original draft preparation, M.M.-Ś. and K.A.-M.; writing—review and editing, M.M.-Ś., K.A.-M. and R.S.; visualization, M.M.-Ś.; supervision, M.M.-Ś. and R.S.; project administration, M.M.-Ś.; funding acquisition, M.M.-Ś. and A.B.-M. All authors have read and agreed to the published version of the manuscript.

**Funding:** This research was funded by the state budget under the Ministry of Education and Science program titled: II International Scientific Conference 'Spatial Management and Natural Resources: project number DNK/SP/546699/2022, dated 21.11.2022, task No. 3, subsidized amount. . . , total project value 143,440.0; RKŁ/DN/2/2024.

**Institutional Review Board Statement:** Not applicable.

**Informed Consent Statement:** Not applicable.

**Data Availability Statement:** https://storymaps.arcgis.com/stories/4f63a83d47e2479eb682b82eddeb4 7d6 (accessed on 20 December 2023).

**Conflicts of Interest:** The authors declare no conflicts of interest.

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
