# Peer review of "Green Roofs: Nature-Based Solution or Forced Substitute for Biologically Active Areas? A Case Study of Lublin City, Poland"

_sustainability, doi:10.3390/su16083131_

Round 1

Reviewer 1 Report

Comments and Suggestions for Authors

Green roofs are often considered as an important construction measure that can achieve urban greening and ecological environment protection. This study selected the ninth largest city in Poland as a case study area, and used field investigation, planning document information analysis and literature review methods to focus on whether green roofs help maintain biodiversity. The manuscript is well organized, the ideas are clear, the methods are reasonable, and the conclusions are interesting. In particular, it was found that not all green roofs fully meet the NBS standards, which is of great reference significance for practical management and policy formulation. However, there are some questions worth discussing before the manuscript is published.

Major comments:

The study area lacks an introduction to the background information on biodiversity.

The authors give a very systematic explanation of the reasons for the findings. But there is a lack of discussion and comparison with existing research, especially that not all green roofs can achieve the purpose of NBS.

Some research methods are still to be refined. Is the Bai index proposed by authors themself or based on existing research? It is recommended to add relevant references. In addition, why is Gr in the formula multiplied by 1/2? How reasonable and recognized is this formula?

Minor comments:

Lines 32-34. Loss of references.

Line 47. change ‘observe’ to ‘observed’.

Lines 326-329. Loss of references.

Comments on the Quality of English Language

Minor editing of English language required

Author Response

Dear Reviewer,

We sincerely appreciate your time and effort in reviewing our scientific article. Your insightful comments regarding the errors and inconsistencies that inadvertently slipped through during the manuscript preparation process are immensely valuable to us. We are committed to addressing each of these points and incorporating necessary revisions into the text. Your feedback will undoubtedly enhance the clarity and accuracy of our work.

Once again, we express our gratitude for your thorough review and constructive criticism. Your input contributes significantly to the refinement of our research, and we look forward to sharing the improved version of our manuscript with you.

The study area lacks an introduction to the background information on biodiversity

Of course, we agree with this comment. When analyzing green roofs from the perspective of promoting biodiversity growth, it's essential to consider its relevance within the study area. We have added a paragraph addressing this in Section 2.1 - study area.

The authors give a very systematic explanation of the reasons for the findings. But there is a lack of discussion and comparison with existing research, especially that not all green roofs can achieve the purpose of NBS.

Thank you very much, we agree with the commends. We will add information in the discussion which more underline the comparison results and existing research.  It seems to be important explain that all of the green roofs which were built in Lublin were taken under consideration. None of them was dedicated as increased biodiversity metrics but require biologically active areas, which percent is regulate by local low or agreements dedicated for the investment.

Some research methods are still to be refined. Is the Bai index proposed by authors themself or based on existing research? It is recommended to add relevant references. In addition, why is Gr in the formula multiplied by 1/2? How reasonable and recognized is this formula?

Thank you very much for pointing out the lack of explanation regarding the presented formula. The Bai index was developed based on provisions of Polish law, specifically on the Regulation of The Minister of Infrastructure dated April 12, 2002, concerning the technical conditions that buildings and their locations should meet [https://isap.sejm.gov.pl/isap.nsf/DocDetails.xsp?id=wdu20020750690, accessed on 25 marc 2024]. These regulations were updated in 2022. The definition of biologically active areas contained in these regulations implies that half of the surface area of a green roof, i.e., a roof providing natural plant vegetation, is considered biologically active. This formula is commonly used to determine the biologically active area index for built-up areas, which is incorporated into the provisions of local spatial development plans. In local spatial development plans in Poland, the required value of the indicator is provided in percentage. 

The necessary information regarding the formula and its application has been added to Section 2.3.

Lines 32-34. Loss of references.

The references have been fulfilled

Line 47. change ‘observe’ to ‘observed’.

Corrected in the text

Lines 326-329. Loss of references.

The references have been fulfilled

Reviewer 2 Report

Comments and Suggestions for Authors

Generally, this paper is very interesting, after a round of revision, it can be accepted.

1. The novelty should be highlighted in Abstract.

2. More Keywords can be added.

3. The main contribution should be summarized in Introduction.

4. What's the research blank of this area? What's the novelty of this paper?

5. References should be updated. One most related reference should be cited.

Random crowd-induced vibration in footbridge and adaptive control using semi-active TMD including crowd-structure interaction. Engineering Structures, 2024, 306: 117839.

6. More sentences should be added in section 3 results.

7. More discussions can be added.

8. Key points of conclusions should be summarized.

Comments on the Quality of English Language

revise

Author Response

Dear Reviewer,

We would like to express our sincere gratitude for conducting the review of our manuscript. We greatly appreciate the insightful comments you provided, which we carefully considered in revising the manuscript. Your feedback has been invaluable in improving the quality of our work, and we have made revisions to address the points you raised.

  1. The novelty should be highlighted in Abstract.

We have taken the comment into account and highlighted the novelty in the abstract.

  1. More Keywords can be added.

We added two additional keywords

  1. The main contribution should be summarized in Introduction.

Thank you very much for the important suggestion. The main and general result important from the studied problem was added in the introduction. 

  1. What's the research blank of this area? What's the novelty of this paper?

Thank you for the question. Our research show and prove deficient of polish local low and agreements dedicated for the investment, which require biologically active areas with out verification of its quantity, nether for biodiversity (f.ex. using BAF) nor for social aspects, (important as classification as an NBS) The lack is used by most the investors who realize green roofs of low quality and low costs of investment, fulfilling formalities and using the green roof as a marking attractions for the clients (commercial and apartment building).  

The information was added both in introduction and discussion. 

  1. References should be updated. One most related reference should be cited.

Thank you for the suggestion regarding citing in our text: Random crowd-induced vibration in footbridge and adaptive control using semi-active TMD including crowd-structure interaction. Engineering Structures, 2024, 306: 117839.

We have reviewed the publication, but unfortunately, we couldn't find a thread that would encompass the research presented in it.

  1. More sentences should be added in section 3 results.

Thank you for the comment. We have expanded the results section

  1. More discussions can be added.

Thank you very much for the important suggestion. More infarction were presents in discussion verifying results of the research.  

  1. Key points of conclusions should be summarized.

Thank you for this remark. We have listed the most important conclusions as suggested.

Reviewer 3 Report

Comments and Suggestions for Authors

The theme of this study is an intriguing topic, but the research content still needs further improvement. The reviewer's comments are as follows:

1. How was the formula for the Biologically Active Areas Index (BAI) established? Was it derived from previous scholars' formulas or created by the authors themselves? This needs to be clearly explained in the paper. If it is based on others' work, proper citation is required; if it is a newly developed formula, the rationale behind its construction needs to be elucidated.

2. Overall, the content of this paper lacks the detailed quantitative calculations and in-depth theoretical analysis expected in academic papers, making it seem more like an outline of a survey report. The main content is merely based on a small sample data to illustrate some phenomena. It overall lacks the rigorous statistical and theoretical analysis that research papers should possess. It is recommended that the authors enhance the paper in terms of sample size selection, spatial variability analysis of evaluation results, mechanism and corresponding countermeasures.

I have evaluated this paper based on my professional background, and my specific comments are as follows:  1. The calculation formula for the Biologically Active Areas Index (BAI) in the text has an unclear source. I suggest the authors provide references. If the formula was developed by the authors themselves, it is recommended that they explain the rationale behind the formula.  2. As an academic paper, this article lacks strong academic rigor. The content of the study reads more like a research report, lacking the theoretical depth expected in an academic paper.  3. The sample size on which this study is based is too small, leading to a certain degree of uncertainty in the conclusions drawn. I recommend that the authors supplement the sample data to enhance the credibility of the research findings.

Author Response

Dear Reviewer,

We would like to express our gratitude for taking the time to review our manuscript. Your insights have been invaluable in helping us improve our work. We acknowledge that your comments are well-founded and warrant further elaboration or expansion of certain content areas. Rest assured, we will carefully consider and incorporate them into the revision of our manuscript.

  1. The calculation formula for the Biologically Active Areas Index (BAI) in the text has an unclear source. I suggest the authors provide references. If the formula was developed by the authors themselves, it is recommended that they explain the rationale behind the formula.  

Thank you very much for pointing out the lack of explanation regarding the presented formula. The BAI index was developed based on provisions of Polish law, specifically on the Regulation of The Minister of Infrastructure dated April 12, 2002, concerning the technical conditions that buildings and their locations should meet [https://isap.sejm.gov.pl/isap.nsf/DocDetails.xsp?id=wdu20020750690, accessed on 25 march 2024]. These regulations were updated in 2022. The definition of biologically active areas contained in these regulations implies that half of the surface area of a green roof, i.e., a roof providing natural plant vegetation, is considered biologically active. This formula is commonly used to determine the biologically active area index for built-up areas, which is incorporated into the provisions of local spatial development plans. In local spatial development plans in Poland, the required value of the indicator is provided in percentage. 

The necessary information regarding the formula and its application has been added to Section 2.3.

  1. As an academic paper, this article lacks strong academic rigor. The content of the study reads more like a research report, lacking the theoretical depth expected in an academic paper.  

Thank you very much for pointing out the structure of the results presented in our research. Indeed, in some aspects, we may have applied excessive mental shortcuts, which could call into question the obtained results. We have supplemented the content with theoretical considerations and expanded the discussion section to verify our findings with other studies. However, increasing the sample size was not feasible. We have analyzed all green-roofed buildings in Lublin, but unfortunately, there were only 13 of them. This result seems particularly intriguing as it demonstrates a trend that does not address the needs related to climate change adaptation actions in cities. Additionally, as highlighted in the title, this is a case study of the city. In the future, we plan to compare our findings with other research areas in Poland and Europe, and perhaps our results will also be utilized by other researchers. As the reviewer rightfully noted, the sample size is small, hence statistical analyses may be unjustified.

  1. The sample size on which this study is based is too small, leading to a certain degree of uncertainty in the conclusions drawn. I recommend that the authors supplement the sample data to enhance the credibility of the research findings.

Thank you very much to the reviewer for the feedback, and we agree that the sample size examined is not very extensive. However, in our study, we aimed to present a case study of green roofs in Lublin. During the preparatory work, we surveyed 13 buildings with roofs or ceilings covered with vegetation. This means that all objects in the city containing green roofs were examined, and the fact that there were only 13 of them speaks solely to the continued limited popularity of such solutions. We do not have the possibility to increase the sample size because there are no more green roofs available.

Reviewer 4 Report

Comments and Suggestions for Authors

Summary

The study examined the extent to which green roofs in the city of Lublin in Poland could be assessed as Nature Based Solutions (NBS) or whether they amounted to little more than greenwashing – adherence to the building codes without going any further. The green substrate for the 13 buildings assessed in the study classified them intensive (substrate >25 cm), extensive (substrate 5 - 15 cm), and semi-intensive (wide variety of types of vegetation with substrate >25 cm). Out of 13 buildings examined, only three fulfilled NBS and these were the only buildings classified semi-intensive.

Assessment

The study used a rather simple gauge of effectiveness – whether the green roofs were intensive, extensive or semi-intensive. It would have been useful to be more specific about the forms of vegetation that comprised the green roof – species, size, density, and also how well the green roofs were managed (or not). The analysis provided is rather minimal.

The material in the Discussion section on the application of green roofs in other European cities would be better placed in the introductory section. The report should identify measures that would improve the green roofs in Lublin.

The method is sound and reproducible. However, its weakness is that it is rather minimal in terms of its analysis and findings. A strength of the paper is the survey of the existing green roofs in Lublin. All but one of the 67 references were post 2000. Given the extensive literature it is surprising that the study did not examine a wider range of issues associated with green roofs.

The figures and tables in the paper are all legible and communicate well.

The authors declared no conflict of interest and provide access to the data used in the study.

Author Response

Dear Reviewer,

We would like to express our sincere gratitude for thoroughly reviewing our work and providing valuable feedback. Your insightful comments have helped us identify the inconsistencies in our manuscript. We carefully considered all your suggestions and incorporated them into the revised text. Additionally, we have clarified our approach to addressing these points in the sections.

Round 2

Reviewer 2 Report

Comments and Suggestions for Authors

Accept 

Comments on the Quality of English Language

revise

Author Response

Dear Reviewer,

We sincerely appreciate your attention to the linguistic quality of our manuscript and your constructive comments regarding the English language used in our research presentation. Following your advice, we have enlisted the assistance of a professional proofreader to thoroughly review and refine the language used in our manuscript.

Enclosed with this response, please find the revised manuscript, which has undergone extensive language corrections to ensure clarity, precision, and academic rigor. We believe these revisions have significantly improved the readability and overall quality of our work.

We are grateful for your insightful feedback, which has contributed to enhancing the presentation of our research. We look forward to any further suggestions you may have and hope our revised manuscript meets your expectations and the high standards of Sustainable journal.

Thank you once again for your invaluable contribution to the improvement of our work.

Reviewer 3 Report

Comments and Suggestions for Authors

The revised manuscript has provided detailed explanations for each of the reviewer's comments. However, there are still several issues in the text that need further refinement: 1. The usage of the unit 'square meters' in the manuscript is incorrect in several places, it is suggested to review the entire text and make corrections.

2. The resolution of the images in the manuscript is low, such as in Figures 1 and 2, it is recommended to provide higher quality images.

3. In the 'Discussion' section of the manuscript, it is advisable to appropriately supplement the uncertainties of conclusions derived from small-sample studies and identify areas for future research improvement.

Author Response

Dear Reviewer,

Thank you very much for your insightful comments and the valuable feedback provided on our manuscript. Your attention to detail has significantly contributed to enhancing the accuracy and completeness of our work.

We have carefully reviewed the text and corrected all the units of measure as per your suggestions. We agree that precision in such details is crucial for the scientific integrity and clarity of our article.

Additionally, we have addressed the issue with the figures that lost their resolution during the PDF generation process. Enclosed with this response, you will find the revised figures with improved resolution, ensuring that they now meet the publication standards and effectively support the research findings presented.

In response to your comments regarding the discussion section, we have also added a paragraph that you rightly pointed out was missing. This new section aims to further elaborate on the implications of our findings and contributes to a more comprehensive understanding of the research context.

We are confident that these revisions have significantly improved the quality of our manuscript. We have attached the updated manuscript for your review and kindly ask for your feedback on the changes made.

Thank you once again for your constructive criticism and for helping us to improve the quality of our submission. We look forward to your further suggestions and hope that our revised manuscript meets your approval as well as the journal’s standards.

Reviewer 4 Report

Comments and Suggestions for Authors

Great improvement over earlier paper

Author Response

Dear Reviewer,

We sincerely thank you for the considerable time and effort you dedicated to reviewing our manuscript. Your detailed and insightful feedback has been invaluable, guiding us to enhance the quality and clarity of our work significantly. We deeply appreciate your commitment to improving academic discourse and have carefully addressed each of your suggestions in our revised submission, attached herewith for your review. We are hopeful that our revisions meet the esteemed standards of Journal and look forward to any further advice you may have.

Warmest regards